# Secretory Profile Analysis of Human Granulosa Cell Line Following Gonadotropin Stimulation

**DOI:** 10.3390/ijms26094108

**Published:** 2025-04-25

**Authors:** Francesca Mancini, Emanuela Teveroni, Michela Cicchinelli, Federica Iavarone, Anna Laura Astorri, Giuseppe Maulucci, Cassandra Serantoni, Duaa Hatem, Daniela Gallo, Carla Di Nardo, Andrea Urbani, Alfredo Pontecorvi, Domenico Milardi, Fiorella Di Nicuolo

**Affiliations:** 1International Scientific Institute Paul VI, Università Cattolica del Sacro Cuore, Fondazione Policlinico Universitario A. Gemelli IRCCS, 00168 Rome, Italy; mancini.chicca@gmail.com (F.M.); emanuela.teveroni@guest.policlinicogemelli.it (E.T.); annalaura.astorri@policlinicogemelli.it (A.L.A.); fiorella.dinicuolo@gmail.com (F.D.N.); 2Department of Basic Biotechnological Sciences, Intensivological and Perioperative Clinics, Università Cattolica del Sacro Cuore, 00168 Rome, Italy; michela.cicchinelli@unicatt.it (M.C.); federica.iavarone@unicatt.it (F.I.); andrea.urbani@policlinicogemelli.it (A.U.); 3Clinical Chemistry, Biochemistry and Molecular Biology Operations (UOC), Fondazione Policlinico Universitario A. Gemelli IRCCS, 00168 Rome, Italy; 4Department of Neuroscience, Section of Biophysics, Università Cattolica del Sacro Cuore, 00168 Rome, Italy; giuseppe.maulucci@unicatt.it (G.M.); cassandra.serantoni@unicatt.it (C.S.); doaahatem44@gmail.com (D.H.); 5Fondazione Policlinico Universitario A. Gemelli IRCCS, 00168 Rome, Italy; 6Dipartimento Scienze della Salute della Donna, del Bambino e di Sanità Pubblica, Fondazione Policlinico Universitario A. Gemelli IRCCS, Lgo A. Gemelli 8, 00168 Roma, Italy; danielagallo65@gmail.com; 7Division of Endocrinology, Fondazione Policlinico Universitario A. Gemelli IRCCS, 00168 Rome, Italy; carla.dinardo@guest.policlinicogemelli.it (C.D.N.); alfredo.pontecorvi@policlinicogemelli.it (A.P.)

**Keywords:** KGN, granulosa cells, secretomic analysis, FSH, hCG, semaphorin

## Abstract

Granulosa cell (GC) differentiation, stimulated by FSH and LH, drives oocyte maturation and follicle development. FSH promotes GC proliferation, and LH triggers ovulation. In clinical practice, hCG is used to mimic LH. Despite various controlled ovarian stimulation (COS) protocols employing exogenous gonadotropins and GnRH analogs to prevent premature ovulation, their effectiveness and safety remain debated. To identify markers predicting a positive treatment response, the secretome of gonadotropin-stimulated GC using the human granulosa-like tumor cell line (KGN) via proteomics was analyzed. Additionally, a novel 2D-FFT quantitative method was employed to assess cytoskeleton fiber aggregation and polymerization, which are critical processes for GC differentiation. Furthermore, the activation of key kinases, focal adhesion kinase (FAK), and Rho-associated coiled-coil-containing protein kinase 1 (ROCK-1), which are implicated in cytoskeleton dynamics and hormone signaling, was evaluated. The proteomic analysis revealed significant modulation of proteins involved in extracellular matrix organization, steroidogenesis, and cytoskeleton remodeling. Notably, the combined FSH/hCG treatment led to a dynamic upregulation of the semaphorin pathway, specifically semaphorin 7A. Finally, a significant reorganization of the cytoskeleton network and signaling was detected. These findings enhance our understanding of folliculogenesis and suggest potential novel molecular markers for predicting patient responses to gonadotropin stimulation.

## 1. Introduction

In mammals, gonadotropin stimulation drives the differentiation of ovarian granulosa cells (GCs), which is essential for oocyte development and follicle maturation [1]. While the precise mechanisms remain incompletely understood, follicle-stimulating hormone (FSH) and luteinizing hormone (LH), secreted by the anterior pituitary gland, play crucial roles [1]. FSH initiates and stimulates GC proliferation, driving the development of preovulatory follicles, while the LH surge triggers follicle rupture and ovulation [1,2,3]. The FSH receptor (FSHR) expression on GC serves as a marker of folliculogenesis [4]. The FSH-FSHR interaction initiates intracellular signaling cascades that lead to steroidogenesis, producing hormones like estradiol and progesterone, which modulate GC proliferation and differentiation [5]. Similarly, LH binding to its G-protein-coupled receptor (LHCGR) elevates the intracellular cAMP levels, activating protein kinase A (PKA) and, subsequently, phosphorylating p42/44 MAPK [6]. This intracellular signaling cascade enhances the expression of genes associated with cellular differentiation, ovulation, and steroidogenesis. Furthermore, it induces post-transcriptional regulation of proteins involved in luteal function maintenance, such as lipolysis, which is essential for providing cholesterol for progesterone synthesis [7,8].

While FSH and LH are the naturally occurring final follicular maturation triggers, human chorionic gonadotropin (hCG) is commonly used in clinical practice during stimulated cycles, as it pharmacologically mimics the effect of LH by binding to the LH receptor with higher affinity, a much longer half-life, and increased bioactivity [9,10].

Various controlled ovarian stimulation (COS) protocols have been developed over the years, but there is no clear consensus regarding their optimal efficacy and safety. This includes debates about the use of recombinant versus extractive gonadotropins and high- versus low-dose regimens. All COS protocols (long agonist, antagonist, or flare protocols) involve administering exogenous gonadotropins to induce multiple ovarian follicle development, alongside drugs that prevent premature ovulation by suppressing the pituitary gland (e.g., GnRH agonists, GnRH antagonists) or by progestins [11,12]. Commonly used gonadotropins include recombinant follicle-stimulating hormone (rFSH), urinary follicle-stimulating hormone (uFSH), and highly purified human menopausal gonadotropin (hp-hMG). rFSH is produced using recombinant DNA technology: follitropin-α and -β are derived from Chinese hamster ovary (CHO) cells, while follitropin-δ is derived from a human fetal retinal cell line [13]. Urofollitropin (uFSH) is a highly purified FSH preparation extracted from human urine. hp-hMG is a mixture of FSH, LH, and hCG, extracted and purified from the urine of post-menopausal women [14]. The final step of follicular development is triggered by hCG, which enables the process of oocyte maturation [15].

Although many studies and meta-analyses have evaluated the efficacy of COS protocols, there is currently no strong biological or clinical evidence to justify the choice of one protocol over the other in terms of ovulation rates and pregnancies [16,17]. Therefore, it would be necessary to highlight “in vitro” a pattern of protein expression modulated by gonadotropins to identify markers of positive response to treatment.

The primary aim of this study was to characterize the secretome of granulosa cells using a human granulosa-like tumor cell line (KGN) following gonadotropin stimulation via a proteomic approach. Furthermore, given the pivotal role of cytoskeleton remodeling in granulosa cell differentiation and its mediation of regulatory protein trafficking to the oocyte [18], a novel quantitative method, Fast Fourier Transform (FFT), to assess protein aggregation and polymerization of cytoskeleton fibers following gonadotropin stimulation, was utilized. Finally, to analyze the early events of the hormone signaling cascade during gonadotropin-induced structural cytoskeleton changes, the activation of crucial kinases, such as focal adhesion kinases (FAKs) and Rho-associated protein kinases (ROCK-1 and -2), was evaluated. FAK is activated in granulosa cells during the ovulation process, particularly through the fibronectin-integrin pathway [19]. This activation is crucial for the morphological changes in GC, leading to progesterone production and successful ovulation [19,20]. As well, ROCKs play an essential role in regulating cytoskeleton dynamics, particularly actin fiber remodeling, which are essential for several cellular processes, such as cell proliferation, adhesion, and mechanical stability [21].

## 2. Results

### 2.1. Secretomic Analysis of KGN Cells upon Gonadotropin Treatments

Differential proteomic analysis based on mass spectrometry revealed quantitative and qualitative differences in the set of proteins secreted into the KGN cell culture medium (secretome) in response to treatment with 1.0 IU/mL FSH, hCG, or the FSH/hCG combination for 48 h.

Figure 1a–c show the Venn diagrams of secreted proteins in each condition with respect to the control (untreated medium (CTR)). In particular, upon the FSH treatment, 273 common proteins were found, 348 were found in the CTR medium, and 43 were uniquely secreted after the gonadotropin treatment. Upon the hCG treatment, 168 proteins were shown in the control medium, 453 were common, and 72 were uniquely secreted. The combination of FSH and hCG showed 433 common proteins and 72 uniquely secreted compared to the 188 found in the CTR medium. Appendix A lists all of the proteins found. The clustergram (Figure 1d) displays a hierarchical clustering analysis of differentially secreted proteins following each gonadotropin treatment (FSH, hCG, or FSH/hCG), revealing significant modulation of protein clusters under different conditions.

In Figure 2a–c, the STRING-generated interaction network of the exclusively secreted proteins belonging to each gonadotropin treatment (the intensity of the edges reflects the strength of the interaction score) is reported.

The analysis of proteins uniquely secreted by the FSH treatments showed several cytoskeleton- or matrix-associated proteins (Figure 2a). In particular, syndecan-4, vitronectin, fibronectin-1 (FN1), and pregnancy zone protein (PZP) were noticed. Syndecans are heparan sulfate proteoglycans that influence the actin cytoskeleton organization and play multiple structural and signaling roles [22]. Vitronectin is a multi-domain matrix-associated glycoprotein, essential for supporting cellular functions such as proliferation, adhesion, and differentiation [23]. Fibronectin, a glycoprotein component of the extracellular matrix (ECM), plays a vital role in cellular processes such as growth, migration, and differentiation. This protein interacts with cell membrane integrins and other ECM components, including heparan sulfate proteoglycans (e.g., syndecans), fibrin, and collagen [24]. Pregnancy zone protein (PZP), an immunosuppressive protein, contributes to extracellular protein homeostasis by engaging with diverse receptors and also functions to suppress T-cell activity during pregnancy [25].

Figure 2b shows the analysis of secreted proteins after treatment with hCG. Among these proteins were found some proteins already observed in the FSH treatment (FN1 and PZP) and others that are specific to this treatment. In particular, superoxide dismutase 2 (SOD2), which is an important factor in apoptotic signaling and oxidative stress [26], and semaphorin 3C, a secretory semaphorin known to affect cytoskeletal remodeling and integrin-dependent cell signaling [27]. Semaphorins play a crucial role in cell–cell communication, particularly in the maturation of the cumulus-oocyte [28]. Interestingly, it has been reported that several semaphorin3a mutations and polymorphisms were associated with infertility [28].

As shown in Figure 2c, KGN granulosa cells treated with an FSH/hCG combination secreted, as expected, several proteins common to both individual treatments (FN1 and SOD2), as well as proteins secreted specifically under the combined treatment. Notably, apolipoprotein A1 (APOA1), which is involved in cholesterol transport and in the integrin-mediated signaling pathway [29], microtubule-associated protein RP/EB family member 1 (MAPRE1), and tropomyosin alpha-1 chain (TPM1). MAPRE1 is involved in two key processes: the dynamic regulation of microtubule structures and the maintenance of chromosome stability [30]. TPM1, an actin-binding protein, plays a role in the contractile system of striated and smooth muscles. Furthermore, it contributes to the cytoskeleton assembly in non-muscle cells [31].

Furthermore, a secretomic analysis revealed 34 significantly upregulated and 44 downregulated proteins upon the FSH treatment (fold change, FC ≥ 1.5-fold of induction or inhibition compared to the CTR-untreated cells). The hCG treatment showed 25 upregulated proteins and 27 downregulated proteins, while the combination caused an upregulation of 25 proteins and a downregulation of 26 proteins. Table 1a–c lists all of the proteins analyzed with the corresponding fold change compared to the CTR-untreated group.

Of note, upon each treatment, there were upregulated proteins involved in steroidogenesis, gluconeogenesis, and extracellular matrix remodeling and signaling, such as apolipoprotein B and E, insulin-like growth factor-binding protein 4, laminin, fibulin, and matrilin-2. Importantly, upon the combined treatment, semaphorin 7A and its receptor integrin 1β were upregulated. Semaphorin 7A exhibits pleiotropic functions ranging from axon growth, angiogenesis, cell growth, and differentiation [32]. Furthermore, dysregulation of semaphorin 7A/β1-integrin signaling has been shown to play an important role in the reproductive process, leading to abnormal gonadal development or impaired fertility [33].

To validate the proteomic data of semaphorin 7A secretion upon the FSH/hCG combined treatment, the levels of this protein in kGN supernatants were evaluated by an ELISA assay. Figure 3a shows a significant increase in semaphorin 7A levels compared to the untreated CTR medium. This result was confirmed by a Western blot analysis (Figure 3b).

Among the downregulated proteins, cofilin, moesin, and vinculin are to be pointed out, which are involved in cytoskeleton remodeling, as well as neuropilin-2, a semaphorin receptor, and HSP90 chaperone for steroid hormone receptors. In the combined treatment, secernin-1 and serglycin, both involved in the formation and function of secretory granules [34,35], were highlighted.

Table 2, Table 3 and Table 4 show a pathway analysis of up- or downregulated proteins by the gonadotropin treatment. 

Among the pathways upregulated (Table 2a) by the FSH treatment, extracellular matrix organization and degradation, MET signaling, PDGF signaling, and the regulation of insulin-like growth factor (IGF) transport and uptake by insulin-like growth factor binding proteins (IGFBPs) were highlighted. MET is a receptor tyrosine kinase (RTK) whose extracellular region consists of three domains: semaphorin (SEMA), plexin-semaphorin-integrin (PSI), and four immunoglobulin-plexin-transcript (IPT) domains. It is able to activate RAS-MAPK signaling, resulting in the initiation of downstream signaling cascades, which ultimately promote cell motility and proliferation [36]. PDGF, by binding to its receptor, PDGFR, activates several signaling pathways, such as the mitogen-activated protein kinase/extracellular signal-regulated kinase (MAPK/ERK) and the phosphatidylinositol 3 kinase protein kinase B (PI3K/AKT/PKB) pathway, promoting cell proliferation and invasion [37]. Insulin-like growth factors (IGFs) are potent mitogens that stimulate cell proliferation. These proteins exist in extracellular fluids bound to specific high-affinity IGF-binding proteins (IGFBPs). Notably, IGFBPs can also modulate signaling pathways at the cell surface through other receptor systems, such as integrins and transforming growth factor β family receptors [38]. Pathways downregulated (Table 2b) by the FSH treatment include glucose metabolism and gluconeogenesis, platelet degranulation, activation and signaling, the IL-12 family, and ATF-6 signaling. It is known that multiple signaling pathways that influence cell metabolism pathways, such as glucose metabolism and gluconeogenesis, are also involved in the control of proliferation and differentiation [39]. In particular, the PI3K/AKT/mTOR pathway, which promotes cell proliferation, regulates glycolysis and glucose import through the modulation of glucose transporters and glycolytic enzymes expression and/or activity and promotes fatty acid synthesis [39]. The IL-12 cytokine family can be secreted by both immune and non-immune cells [40]. Molecular signaling mechanisms involving IL-12 family members are all mediated by tyrosine phosphorylation of the Janus family kinases (JAK2 and TYK2), which, in turn, phosphorylate and activate STATs (STAT 1-4), resulting in the transcription of target genes that mediate several biological processes, such as cell growth, differentiation, and apoptosis [41,42]. ATF-6, a transmembrane transcription factor sensitive to endoplasmic reticulum (ER) stress, functions within the unfolded protein response (UPR), a mechanism designed to ensure proper protein synthesis and folding [43]. The ER plays a vital role in protein synthesis, folding, transport, calcium homeostasis, steroid synthesis, and lipid metabolism. The accumulation of misfolded proteins in the ER triggers the proteolytic cleavage of ATF6. The resulting cytosolic fragment of ATF6 translocates to the nucleus, where it stimulates the expression of ER chaperones and folding enzymes [44].

Among the pathways upregulated (Table 3a) by the hCG treatment, NR1H1 and NR1H3 signaling and ALK signaling were found, while pathways downregulated (Table 3b) by the hCG treatment include platelet degranulation, activation, and signaling (previously observed in the FSH treatment) and the RHOBTB GTPase cycle. The hepatic X receptors (LXR) NR1H3 and NR1H2 play a key role in cholesterol metabolism [45]. These receptors belong to a subclass of nuclear receptors that form heterodimers with 9-cis retinoic acid receptors (RXRs). Furthermore, they are activated by oxysterols (oxidized cholesterol), whose intracellular levels correlate with cholesterol concentrations [46]. Upon ligand binding (oxysterols), they initiate the transcription of several genes involved in lipid metabolism, particularly HDL-cholesterol metabolism [46]. Anaplastic lymphoma kinase (ALK) is a receptor tyrosine kinase that plays a crucial role in embryonic development and cellular differentiation [47]. ALK signaling led to the activation of several downstream pathways, including the PI3K/AKT, RAS/MAPK, JAK/STAT, CRKL/C3G/RAP1, and PLCγ/DAG/PKC pathways, resulting in enhanced cell proliferation, migration, survival, and angiogenesis [48]. Rho GTPases are key regulators of signaling pathways that control cytoskeletal dynamics and, consequently, all processes that depend on the rearrangement of the actin cytoskeleton, such as cytokinesis, membrane trafficking, cell motility, and adhesion [49]. They also play a role in a variety of processes not directly related to actin reorganization, such as cell cycle progression, NADPH oxidase activation, gene expression, and apoptosis [50].

As shown in Table 4a, the treatment with FSH plus hCG resulted in an upregulation of insulin-like growth factor (IGF) transport and uptake by IGFBPs (previously observed in the FSH treatment) and the semaphorin interaction pathway. Semaphorins, as mentioned above, play crucial roles in various tissues and systems, affecting cytoskeleton organization and signaling through integrin, neuropilin, and plexin receptors [51]. Of note, the modulation of semaphorin interaction and signaling occurs only following the combined treatment, highlighting that the action of both hormones is necessary for complete follicle development. Among the downregulated pathways (Table 4b) upon treatment with a combination of FSH and hCG, platelet degranulation and activation and the RHOBTB GTPase cycle pathway (previously observed in the single treatments) were found.

Comparisons of the secreted proteins (FSH vs. FSH/hCG and hCG vs. FSH/hCG) revealed similarities to the differential protein expression seen when each treatment was compared to the control, as detailed in Appendix A.

### 2.2. KGN Cytoskeleton Analysis upon Gonadotropin Treatments

Since the secretome revealed the modulation of several molecules involved in cytoskeleton remodeling, a key element in mediating the molecular trafficking of regulatory proteins and in granulosa cell differentiation, the effects of FSH and hCG (alone or in combination) on KGN cytoskeleton rearrangement and on molecular signaling involved in this process were investigated. Our results showed that treatment with gonadotropins results in a massive change in F-actin organization, with a remodeling of the actin cytoskeleton. The microscopy images of the KGN cells treated with different gonadotropins are presented in Figure 4a (left panel). The four pictures show immunofluorescence of F-actin filaments (green fluorescence) of the untreated cells (CTR) and after the FSH, hCG, and FSH/hCG treatments. The nuclei stained with DAPI are shown in blue. In Figure 4b (right panel), a 2D-FFT analysis shows changes in cytoskeletal density and aggregation under various treatment conditions. These alterations match with granulosa cell secretomic data that are focused on cytoskeletal dynamic rearrangement. As shown in Figure 4c, confocal microscopy demonstrated that the treatment of KGN cells with FSH (*p* = 0.03), hCG (*p* = 0.03), or FSH/hCG (*p* = 0.04) results in significant changes in F-actin fiber polymerization amplitude compared to the untreated control (CTR). In addition, a Western blot analysis shows a dynamic activation of intracellular signaling involved in the structural alterations of F-actin fibers (Figure 4d). In particular, the phosphorylation of regulatory cytoskeletal protein kinases, such as the Rho kinase (ROCK) and focal adhesion kinase (FAK), was evaluated. As shown, 15 min of treatment with FSH alone or in combination with hCG significantly increased FAK phosphorylation on Tyr^397^ and ROCK phosphorylation on Tyr^913^, suggesting a key role of FSH in this post-translational modification.

## 3. Discussion

A comparative proteomic analysis of KGN culture media was conducted to assess the impact of FSH, hCG, and a combination of FSH and hCG treatments on granulosa cell secretomes.

Clinically, human chorionic gonadotropin (hCG) is a recognized luteinizing hormone (LH) mimetic, commonly employed for ovulation induction and luteal phase support. However, it is crucial to acknowledge that hCG’s action is not entirely equivalent to that of LH. Although both hormones bind to the shared LH/hCG receptor, hCG’s longer half-life and subtle distinctions in downstream signaling can result in different physiological effects compared to the pulsatile release of endogenous LH [52]. Additionally, hCG displays a weak ability to bind and stimulate the follicle-stimulating hormone (FSH) receptor, albeit with considerably lower affinity than FSH [53]. Therefore, the selection between LH and hCG necessitates careful evaluation of the specific objectives of each experiment and the potential impact of their inherent differences. Regardless, the present study investigated the effects of FSH and hCG on KGN cells to identify potential markers indicative of a positive response to gonadotropin therapy by simulating controlled ovarian stimulation (COS) protocols.

The analysis of the uniquely secreted proteins after FSH treatment shows several molecules strictly related to actin cytoskeleton organization that may have an impact on granulosa cell adhesion and differentiation and on the extracellular matrix environment.

The proteoglycans, in particular syndecan-1 and syndecan-4, play an essential role in oocyte growth differentiation factor (GDF)-9 signaling and affect the cumulus cell function in the periovulatory follicle [54]. During the ovarian cycle, syndecan-1 expression in granulosa cells shows cyclic changes, suggesting that it may play specific roles in follicle development [55]. Colombe S. et al. demonstrated that syndecan-1 regulates KGN granulosa cell differentiation and follicular development through a variety of mechanisms involving morphological changes, control of signaling pathways, and alterations in gene expression [55]. Syndecan-4 connects the cytoskeleton with the extracellular matrix, and it is involved in several biological processes, such as cell–matrix adhesion, endocytosis, mechanotransduction, cell migration, and polarity [56]. Moreover, it is able to interact with fibronectin and bind several growth factors such as FGF-2 and myostatin, a TGF-β superfamily member [57,58].

Matrix-associated proteins are essential for supporting cellular functions like proliferation and differentiation and provide a molecular scaffold for signaling molecules, such as integrins and growth factors, by forming an extracellular–intracellular “bridge” for the transmission of signals between different cells separated by the basal lamina [59].

Vitronectin (Vn), a multi-domain glycoprotein, is a component of the extracellular matrix (ECM), along with fibronectin and type I collagen. Vn anchors to the ECM via heparin- or collagen-binding domains, mediating cell adhesion and migration by interacting with integrins (α3β1, αvβ1, αvβ3, αvβ5, and αIIbβ3) [60]. Vn-bound integrins activate signaling pathways that regulate cytoskeletal reorganization, gene expression, and ion transport [61,62]. Matsushige et al. (using a 3D ovarian tissue culture) demonstrated that the Vn RGD domain stimulates theca cell differentiation in ovarian interstitial cells through integrins αvβ3 and αvβ5, promoting antral follicle growth [63]. Among the proteins secreted in response to FSH, it is interesting to highlight PZP, a protease inhibitor that is a major pregnancy-associated plasma protein. Finch et al. showed that plasma PZP levels were higher during pregnancy, and this phenomenon can be considered a maternal adaptation mechanism to maintain extracellular protein homeostasis during human pregnancy [64]. Moreover, PZP is significantly downregulated in pathological conditions such as preeclampsia or HELLP syndrome [65]. Recent research showed that PZP not only plays a role in pregnancy, but it also influences several processes such as lipid metabolism and tumorigenesis [66,67]. In addition, it has been demonstrated that PZP interacts with different receptors or chaperone proteins, including transforming growth factor-β (TGF-β), glycoside A (GdA), and low-density lipoprotein receptor-related protein (LRP) [68,69]. Interestingly, in a comparative study, Yding Andersen and colleagues showed that PZP plasma levels during the follicular phase were significantly higher at the time of oocyte aspiration in women undergoing ovarian stimulation for “in vitro” fertilization–embryo transfer (IVF-ET) treatment compared to unstimulated women [70].

Among the proteins uniquely secreted following hCG treatment was noted semaphorin 3C, a secretory semaphorin known to influence cytoskeletal remodeling and integrin-dependent cell signaling [27]. Okabe et al. found that semaphorin 3C is present in granulosa cells and is secreted into the extracellular matrix within the cumulus and granulosa cell layers [71]. Furthermore, the downregulation of semaphorin C by siRNA dramatically suppressed the cell migration of granulosa cells and the phosphorylation of focal adhesion kinase (FAK), with an increase in the level of phosphorylated ROCK, indicating that Sema3C regulates actin reorganization in a manner dependent on the FAK signaling pathway [71].

Among the proteins uniquely secreted upon a combination of FSH and hCG treatment, ApoA1, MAPRE, and TPM1 were highlighted. ApoA1, a component of high-density lipoprotein (HDL), plays a role in lipid metabolism. In granulosa cells, ApoA1 expression is regulated by the progesterone receptor (PGR), playing a role in follicular growth and angiogenesis [72]. In addition, the Fas/APO-1/CD95 system is crucial for mediating apoptosis in these cells, contributing to follicle atresia [73]. These mechanisms highlight the complex interplay of genetic regulation and apoptotic pathways in ovarian follicle development and regression. MAPRE1 and TPM1 proteins play a crucial role in the regulation of microtubule dynamics and cytoskeletal organization, which are essential for various cellular processes, such as cholesterol synthesis, cell adhesion, and differentiation [74,75].

Following each treatment, several proteins involved in steroidogenesis, gluconeogenesis, and extracellular matrix remodeling and signaling were upregulated. These included apolipoprotein B and E, insulin-like growth factor-binding protein 4, laminin, fibulin, and matrilin-2. Notably, the combined treatment specifically upregulated semaphorin 7A and its receptor, integrin β1. Messina and coworkers showed that the semaphorin 7A/β1-integrin signaling pathway is involved in reproductive processes, and its dysregulation potentially leads to abnormal gonadal development and reduced fertility [33].

The pathway analysis of differential secreted proteins showed that FSH upregulates a cluster of molecules involved in extracellular matrix organization and interactions, cytoskeleton reorganization, and focal adhesion kinase (FAK) signaling. On the other hand, FSH downregulates proteins involved in glucose metabolism and gluconeogenesis, as well as cofilin and the ERM family that organize actin filaments and are crucial for the remodeling of cytoskeleton F-actin and the redistribution of adhesion receptors. hCG treatment led to the upregulation of trans-Golgi network vesicle budding, nuclear receptor subfamily 1 (NR1H2 and NR1H3), and ALK-mediated signaling, important for healthy follicle growth and oocyte maturation [76]. Among the downregulated proteins, vinculin, tubulin, alpha actinin-1, and beta-actin-related proteins were noticed, which play pivotal roles in modulating cell adhesion and differentiation [77]. Interestingly, the combination of FSH and hCG, the final stage for the stimulation of follicular rupture and ovum maturation, showed an upregulation of semaphorins and their receptors as integrin β1. A downregulation of proteins essential for ECM reorganization, including serglycin, was observed. Serglycin’s reduced expression suggests a decrease in the activation of intracellular cascades that control proteolytic enzymes, as well as pathways that enhance invasion, stemness, EMT, and ECM remodeling [78,79,80].

Of note, recently, Tremblay P. and Sirard M. employed RNA-seq technology to analyze gene expression in KGN cells following treatment with either FSH or specific activators of protein kinases A, B, and C (PKA, PKB, and PKC). Even though this study concerns cellular RNA rather than secreted molecules, they found a modulation of the differentiation and steroidogenesis pathways upon FSH treatment, providing additional confidence to our data [81].

Our “in vitro” model showed a notable increase in the expression of two semaphorins: semaphorin 3C upon the hCG treatment and semaphorin 7A with the combined FSH and hCG treatment. Moreover, the semaphorin pathway modulation, particularly after the combined gonadotropin treatment, implies that both hormones are necessary for full follicle development and oocyte maturation.

Semaphorins (Sema) are a large family of extracellular signaling proteins, characterized by a conserved amino-terminal semadomain and classified into five vertebrate subclasses (Sema3-7). Several semaphorins are implicated in regulating crucial aspects of follicle development and steroidogenesis. Specifically, the glycoproteins Sema3A, Sema6A, and Sema6D, involved in cell-to-cell communication, exhibit high expression in ovarian cumulus cells. Furthermore, granulosa cell expression of Sema4D and its receptor plexin B1 can modulate steroid hormone secretion and follicular growth [82,83]. Supporting this, Chen et al. reported that reducing Sema4C levels in mouse primary granulosa or thecal interstitial cells significantly impaired ovarian steroidogenesis and disrupted the actin cytoskeleton. Notably, the cytoskeleton-related RHOA/ROCK1 pathway was also suppressed following Sema4C downregulation [84]. As a member of the semaphorin family, semaphorin 7A (Sema7A) is a cell surface glycoprotein anchored to the cell membrane by a glycosylphosphatidylinositol (GPI) moiety. Under specific conditions, Sema7A can be cleaved by metalloproteinases into soluble Sema7A (sSema7A) [85]. Scott et al. showed that Sema7A interacts with plexin C1, leading to the disruption of cytoskeletal actin filaments and the weakening of cell contacts. Conversely, semaphorin 7A binding with integrins (α1β1 or α5β1) promotes cell attachment [86]. Recently, Emery et al. showed that semaphorin 7A expression in mouse preovulatory follicles is mediated by gonadotropins and hypoxia, resulting in a Sema7A asymmetrical expression pattern that is enriched at the apex of the large antral follicles. Moreover, expression of Sema7A was downregulated in the periovulatory phase through a PGR-dependent mechanism, and the asymmetric pattern appears more homogeneous after an ovulatory stimulus [87]. As the role of semaphorin 7A in the regulation of cell-ECM contact and tissue organization is known [88], the dynamic hormonal modulation of Sema7A and its different spatial patterning potentially contribute to follicle structuring during ovulation and luteinization phases. In our “in vitro” model, semaphorin 7A was secreted by KGN granulosa cells upon FSH/hCG combined treatment. This result, although it seems to be in contrast with Emery’s group data, fits with the significant cytoskeleton rearrangement and with the increase in FAK phosphorylation observed. Indeed, it has been reported that semaphorin 7A plays a role in cell migration by increasing FAK activation [89].

Remodeling of the cytoskeleton is a hallmark of granulosa differentiation and is characterized by massive reorganization of the F-actin fibers, formation of lamellipodia and filopodia, and spreading of multilayer cell aggregates to monolayers [18]. Furthermore, the cytoskeleton regulates cholesterol storage and trafficking and, thus, can modulate steroidogenesis, including in periovulatory granulosa cells [90,91,92]. In our research, F-actin aggregation and polymerization were quantified using an innovative approach, integrating quantitative measurements, such as fluorescence intensity, amplitude, and eccentricity features [93]. A significant cytoskeleton fiber reorganization following each gonadotropin treatment was observed. In addition, the activation of different molecular pathways involved in gonadotropin-induced structural cytoskeleton changes, such as FAK and ROCK, was shown. It has been reported that FAK is phosphorylated in granulosa cells during the ovulation process, particularly in response to the synergistic action of fibronectin and amphiregulin (AREG). This phosphorylation is crucial for the morphological changes in granulosa cells that lead to progesterone production and successful ovulation [19]. In addition, inhibition of FAK phosphorylation can suppress ovulation and luteinization of granulosa cells, highlighting its importance in these processes [19]. While Chen shows that Sema4C modulates ovarian steroidogenesis through the RHOA/ROCK1-mediated actin cytoskeleton reorganization [80], the role of ROCK signaling in granulosa cells is still being investigated. Although the ROCK pathway is recognized as a key player in various cellular processes, including migration and cytoskeletal dynamics, its specific contribution to follicular development requires further clarification [94].

Overall, the proteomic analysis showed that FSH/hCG treatment significantly altered proteins related to the extracellular matrix, steroidogenesis, and cytoskeleton remodeling. These findings improve our understanding of folliculogenesis and point to potential new markers for predicting how patients will respond to gonadotropin stimulation. Integrating a secretomic analysis with RNA-seq data will be of interest to investigate whether the observed proteomic changes in protein secretion are a consequence of transcriptional regulation, post-translational modifications, or variations in protein stability.

## 4. Materials and Methods

### 4.1. Cell Culture

The KGN human granulosa-like tumor cell line (provided by Prof. Gallo D.) was used. The KGN cells retain the key physiological characteristics of ovarian granulosa cells, including functional FSH receptor expression and steroidogenic activity [95]. The cells were cultured in phenol red-free DMEM:F12 medium supplemented with 10% charcoal-stripped fetal calf serum, L-glutamine, non-essential amino acids, and penicillin/streptomycin at 37 °C in a 5% CO_2_/95% O_2_ atmosphere. The KGN cells were seeded at 80% confluence (triplicates per condition), washed with warm PBS, and serum-starved for 24 h. Subsequently, the cells were treated with 1.0 IU/mL of FSH (Fostimon^®^, IBSA FARMACEUTICI ITALIA SRL, Lodi, Italy), hCG (Gonasi HP^®^ IBSA FARMACEUTICI ITALIA SRL), or a combination of both in serum-free DMEM/F12-containing antibiotics. The 1.0 IU/mL dose was selected due to its established use in several in vitro studies for granulosa cell stimulation [96,97].

After 48 h, the supernatant containing all factors secreted by the cells (secretome) was harvested from the cultures, centrifuged (10 min, 1000× *g*, room temperature (RT)) to eliminate cell debris, aliquoted, and stored at −80 °C.

### 4.2. Enzymatic Digestion and Mass Spectrometry Analysis

For the proteomic profiling, each sample was analyzed in triplicate. Protein digestion was performed using the filter-aided sample preparation (FASP) protocol [98,99]. Briefly, 50 µg of protein from each sample was reduced (8 mM DTT in 8 M urea, 100 mM Tris), alkylated (50 mM IAA in 8 M urea, 100 mM Tris), and digested with trypsin (1 μg/μL) on Microcon^®^ Centrifugal Filter Devices (Merck Millipore, Darmstadt, Germany). A bottom-up proteomic analysis was conducted using an UltiMate™ 3000RSLCnano HPLC system (Thermo Fisher Scientific, Waltham, MA, USA) coupled to an Orbitrap Fusion Lumos Tribrid Mass Spectrometer (Thermo Fisher Scientific) with an ESI source. The peptides were separated on a PepMap RSLC C18 column (2 µM, 100 Å, 50 µm × 15 cm; Thermo Fisher Scientific) using a gradient elution with 0.1% formic acid (FA) in water (eluent A) and 80% acetonitrile (ACN), 20% water, and 0.1% FA (eluent B). The gradient was as follows: 3% B (0–110 min), 20% B (110–120 min), 40% B (120–125 min), 90% B (125–145 min), and 3% B (145–155 min) at a flow rate of 0.300 μL/min. The injection volume was 5 μL (1 μg of peptides). The ion source was NSI, with positive polarity (1800 V) and an ion transfer tube temperature of 275 °C. MS/MS spectra were acquired in the data-dependent scan mode (DDS) using an Orbitrap detector at 120,000 resolution (375–1500 *m*/*z*) with higher-energy collisional dissociation (HCD) fragmentation. The samples were analyzed in triplicate. The MS/MS data were processed using Proteome Discoverer 2.4.1.15 (Thermo Fisher Scientific) with the SEQUEST HT algorithm (University of Washington, Seattle, WA, USA) against the UniProt KB/Swiss-Prot Homo Sapiens database. The parameters included the following: a precursor mass of 350–5000 Da, an S/N threshold of 1.5, a precursor mass tolerance of 10 ppm, a fragment mass tolerance of 0.02 Da, a maximum of 2 missed cleavages, a peptide length of 6–144, a dynamic oxidation of +15.995 Da on M, static carbamidomethyl of +57.021 Da on C, and an FDR at 0.01 (strict) and 0.05 (relaxed). The protein abundance was determined by the LFQ analysis using the top 3 average precursor areas. For the proteomic profiling study, each sample was run in triplicate.

### 4.3. Bioinformatics Analysis

A bioinformatic analysis of the mass spectrometry data from the control (CTR), FSH, hCG, and FSH/hCG samples was conducted. To ensure high confidence, only proteins identified by at least two unique peptides were included in the subsequent analyses. Both qualitative and quantitative investigations were conducted. For the qualitative analysis, the proteins identified in the four samples were considered. The intersection of the identified protein lists was used to construct Venn diagrams, finding exclusive and common proteins among the samples. The following comparisons were evaluated: CTR vs. FSH, CTR vs. hCG, and CTR vs. FSH/hCG. Venn diagrams were generated using the online tool Venn Diagram Generator (https://bioinformatics.psb.ugent.be/webtools/Venn/, accessed on 19 June 2023). For the quantitative analysis, the protein abundances were used to generate a heatmap and a clustered heatmap (clustergram), considering proteins quantified in all the samples. The analysis was performed using MetaboAnalyst 6.0 (https://www.metaboanalyst.ca/MetaboAnalyst/, accessed on 19 June 2023). The data were log-transformed and feature-normalized using auto-scaling. A differential protein expression analysis was performed to assess the changes in protein expression across the samples. A quantile normalization was performed, and then the log2 fold change values were calculated, with the significance threshold set at 1.5. The following comparisons were made: FSH/hCG vs. CTR, FSH vs. CTR, and hCG vs. CTR. The fold change ratios and quantile normalization were obtained using Statistics and Machine Learning Toolbox, MATLAB (2022b release) (The MathWorks, Inc., Natick, MA, USA, 2022). Differentially expressed proteins were selected for a pathway enrichment analysis using the Reactome Pathway database (https://reactome.org/). For each pathway, the following parameters were considered: entities found, total entities, entities *p*-value, entities FDR.

### 4.4. Immunoenzymatic Assay (ELISA)

The KGN supernatants, untreated (CTR) and after 48 h of a combination of FSH and hCG treatment, were collected after centrifugation (1500× *g*, 10 min). Semaphorin-7A levels were assessed using a commercial ELISA kit (Thermo Fisher Scientific, Waltham, MA, USA), according to the manufacturer’s protocol.

### 4.5. Quantitative Analysis of F-Actin Cytoskeleton Fiber Organization

To examine the KGN cell cytoskeleton remodeling, the cells (5 × 10^4^ cells/dish) were seeded on glass-bottom dishes (Ibidi, Munich, Germany) in serum-free DMEM and treated with gonadotropins (CTR, FSH, hCG, and FSH/hCG). Following the treatment, the cells were washed with PBS, fixed with 4% PFA (10 min, room temperature), and permeabilized with 0.1% Triton-X (5 min). The F-actin fibers were then stained with FITC-Phalloidin (Sigma-Aldrich, St. Louis, MO, USA) for 30 min at room temperature, according to the manufacturer’s protocol, and visualized using an inverted confocal microscope (Leica SP2, Wetzlar, Germany) with a 60× oil immersion objective (NA 1.4). To quantify the protein aggregation and polymerization of the cytoskeleton fibers, an innovative approach based on 2D Fast Fourier Transforms (2D-FFT) was used [93,100,101]. This method, by integrating quantitative measures, such as fluorescence intensity, amplitude measurements, and eccentricity, allows for a more comprehensive analysis of cytoskeletal dynamics and enables the detection of subtle changes in cytoskeletal organization compared to traditional qualitative methods.

### 4.6. Western Immunoblotting

FAK and ROCK phosphorylation was investigated by Western immunoblotting (WB). The total lysates were obtained from KGN cells treated with 1.0 IU/mL FSH, hCG, or an FSH/hCG combination (in serum-free DMEM/F12) for 15 min (lysis RIPA buffer: 150 mM NaCl, 50 mM Tris, pH 8.0, 0.25% deoxycholate, 0.1% SDS, and 1% NP-40, with protease inhibitors) and separated by 10% SDS-PAGE electrophoresis under reducing conditions. After the electrophoresis, the proteins were transferred to PVDF membranes (Merck Millipore, Darmstadt, Germany). The membranes were incubated for 30 min at room temperature in blocking buffer (EveryBlot Blocking Buffer, BioRad, Hercules, CA, USA) and then overnight (4 °C) with a specific primary antibody (anti-pFAK antibody, Merck Millipore, Darmstadt, Germany, catalogue number 341292; anti-pROCK antibody, Thermo Fisher, catalogue number PA5-105054).

The membranes were incubated with specific horseradish peroxidase-conjugated HRP-IgG, diluted to 1:5000 in blocking buffer. A bound secondary antibody was detected by chemiluminescence. All of the immuno-complexes were analyzed using enhanced chemiluminescence (ECL™ Amersham, Little Chalfont, Buckinghamshire, UK) by a chemiluminescence imaging system, Alliance 2.7 (UVITEC, Cambridge, UK), and quantified by the software Alliance V_1607. The levels of the target phosphorylated proteins were estimated versus the level of the not phosphorylated (total) protein or versus the constant level of β-tubulin (anti-β-tubulin antibody, Sigma-Aldrich, St. Louis, MO, USA, catalogue number T0198).

To confirm the results obtained by the ELISA, the SEMA7A protein levels in the supernatants of the KGN cells treated with a combination of FSH and hCG for 48 h by WB using a primary anti-SEMA7A antibody (anti-semaphorin7A antibody, Merck, Darmstadt, Germany, catalogue number WH0008482M1) and a specific secondary antibody (horseradish peroxidase-conjugated HRP-IgG diluted 1:5000 in blocking buffer) were evaluated. As described above, the bound secondary antibody was detected by chemiluminescence (chemiluminescence imaging system, Alliance 2.7, UVITEC, Cambridge, UK) and quantified by the software Alliance V_1607.

### 4.7. Statistical Analysis

Quantitative data were presented as means ± standard deviation (SD) or medians with minimum–maximum ranges. A statistical analysis was performed using a one-way ANOVA, followed by Bonferroni’s post hoc test. The results are shown as means ± standard errors (SEs) or medians with confidence intervals. GraphPad Prism 7.0 was used for all statistical analyses, with *p* < 0.05 considered statistically significant.

## 5. Conclusions

Controlled ovarian stimulation (COS) is a critical component of assisted reproductive technologies. It involves the use of hormonal treatments to stimulate the ovaries to produce multiple follicles, thereby increasing the number of oocytes available for fertilization. Various protocols have been developed to optimize outcomes, each with its own mechanisms and applications. COS are diverse and can be tailored to individual patient needs based on ovarian reserve markers and specific clinical scenarios. Continued research and refinement of these protocols are essential to improve outcomes in assisted reproduction. To identify protein markers associated with a positive response to gonadotropin treatment, a proteomic approach to analyze the secretome of KGN cells, a human granulosa-like tumor cell line, following FSH, hCG, and a combined FSH/hCG stimulation was used. A massive reorganization of the cytoskeleton network and signaling that mediates granulosa cell differentiation was found. Of note, a dynamic regulation of the semaphorin pathway has been reported, and, particularly, an upregulation of semaphorin 7A upon the combined FSH/hCG treatment. Although further studies should be carried out to confirm these results in biological fluids of women undergoing a COS protocol, our data will have an impact on both basic biological knowledge and possible clinical applications in the field of female infertility treatments.

## Figures and Tables

**Figure 1 ijms-26-04108-f001:**
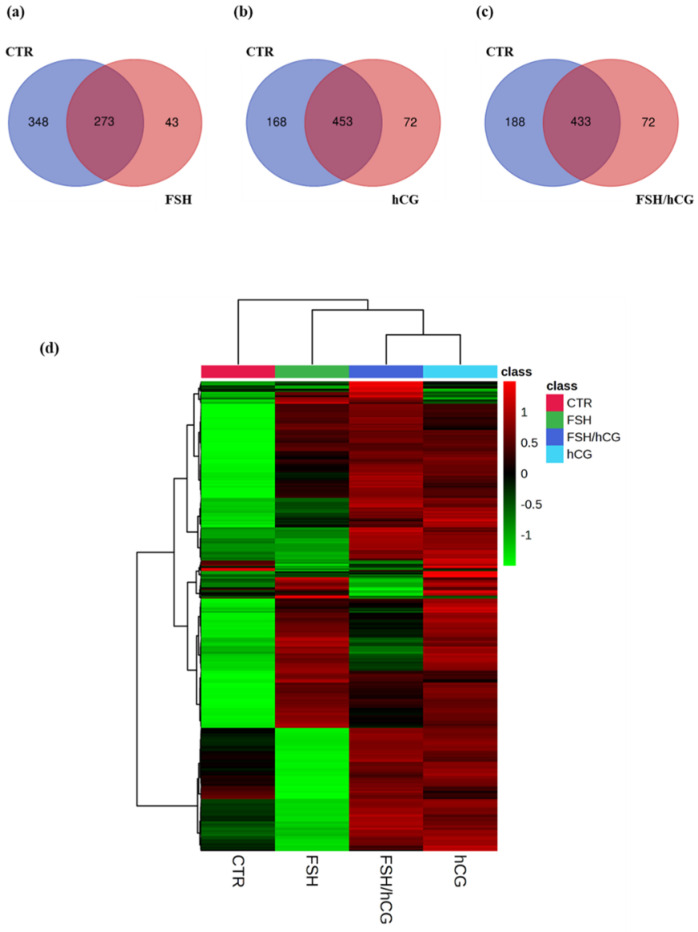
Secretomic analysis Venn diagrams resulting from grouping analysis of the proteins identified in the gonadotropin treatments versus the untreated group (CTR). (**a**) FSH vs. CTR; (**b**) hCG vs. CTR; (**c**) FSH/hCG vs. CTR. The diagrams show the number of proteins identified for each group, with elements shared by both groups, elements belonging exclusively to the gonadotropin treatment, and elements belonging exclusively to the CTR group. (**d**) Clustergram showing the results of the proteomic analysis; the protein levels for each group of treatments are presented in analytical triplicate. The colors represent Z-scores calculated from relative protein expression (z = (relative expression − mean)/standard deviation). Red indicates higher relative expression (high positive Z-score), while green indicates lower relative expression (low negative Z-score). The colors correspond to the Z-score (z = (relative expression − mean)/standard deviation) based on the relative protein expression, with red color (high positive value) indicating higher relative expressions, while green color (low negative value) indicates the lower relative expressions.

**Figure 2 ijms-26-04108-f002:**
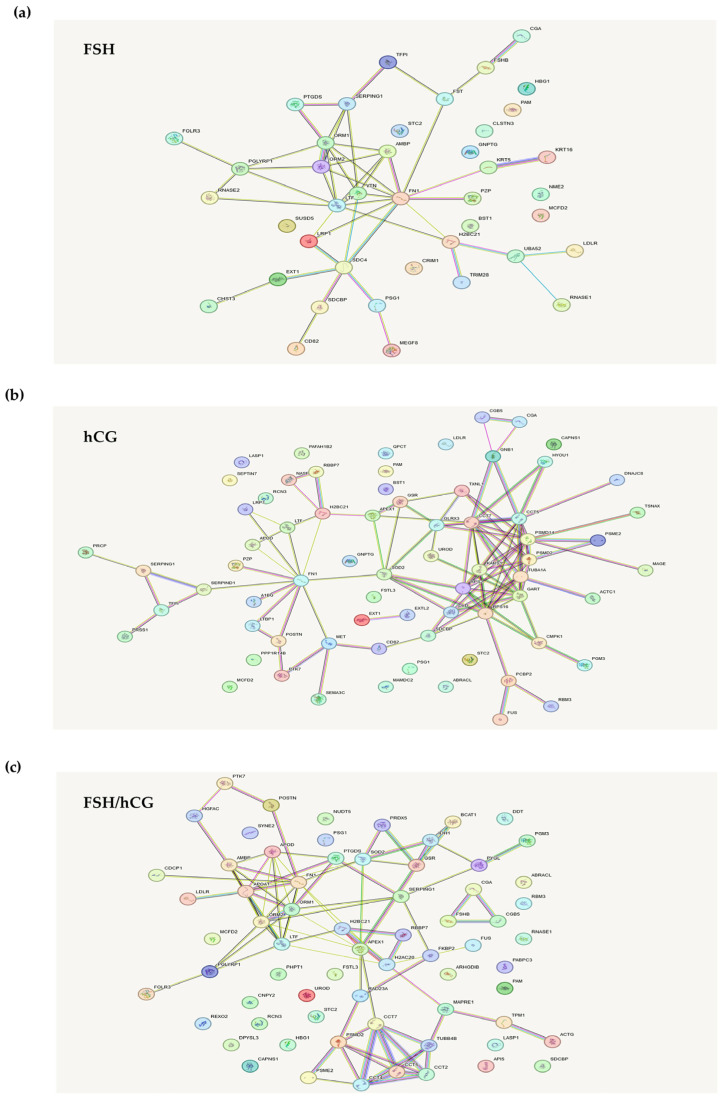
STRING database analysis. STRING analysis shows an interaction network of the exclusively secreted proteins belonging to each gonadotropin treatment: (**a**) FSH; (**b**) hCG; (**c**) FSH/hCG. This network represents protein interactions. Nodes correspond to proteins, and edges indicate interactions. The edges are color-coded to denote seven distinct types of evidence supporting the interactions: red (fusion), green (neighborhood), blue (co-occurrence), purple (experimental), yellow (text mining), light blue (database), and black (co-expression).

**Figure 3 ijms-26-04108-f003:**
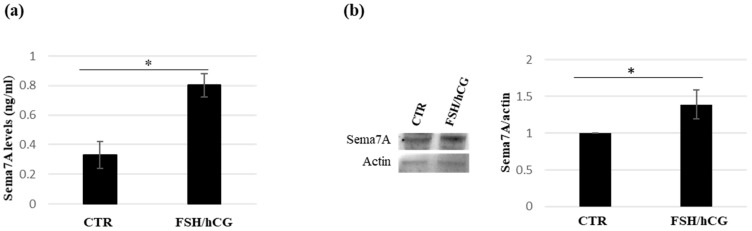
Analysis of semaphorin 7A (Sema7A) levels in KGN medium. (**a**) Immunoenzymatic assay (ELISA) of Sema7A levels in KGN medium upon FSH/hCG combined treatment and in untreated CTR medium. Data are mean ± SD of three independent experiments, * *p* < 0.05. (**b**) Representative Western blot analysis of the Sema7A in KGN medium (CTR vs FSH/hCG treatment). The histogram displays the ratio of Sema7A to actin densitometric values (Sema7A/actin), with actin serving as a loading control. The Sema7A/actin ratio in untreated KGN cells (CTR) is normalized to 1. Data represent the mean ± standard deviation (SD) of three independent biological replicates (* = *p* < 0.05).

**Figure 4 ijms-26-04108-f004:**
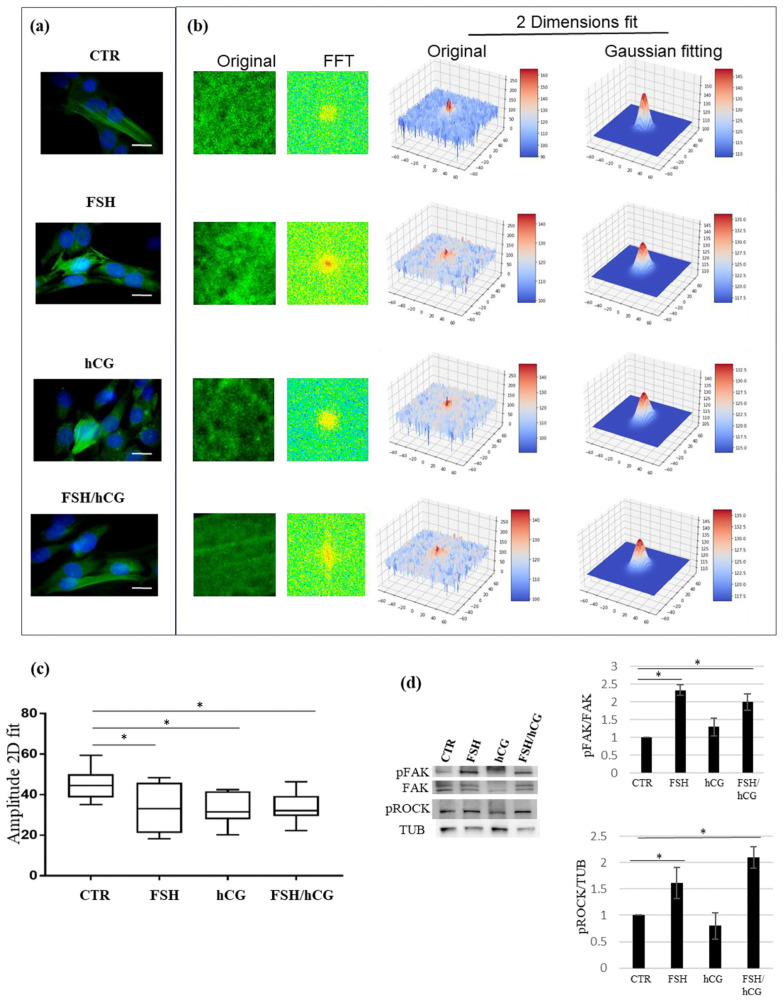
Cytoskeleton analysis. (**a**) Representative pictures of merged confocal immunofluorescence analysis of F-actin staining (green) and nuclei (counterstained with DAPI, blue) in untreated KGN cells (CTR) or upon gonadotropin treatments (FSH, hCG, or FSH/hCG) (left panels). Scale bar: 10 μm. In the right panel (**b**), F-actin polymerization measured by a 2D-FFT analysis is shown. For each image, several squared ROIs (100 × 100 pixels) were selected in correspondence with the inner part of the cells. (**c**) Amplitude analysis showing that KGN cell treatments with FSH, hCG, or with a combination of FSH and hCG significantly affect F-actin polymerization and cytoskeleton spatial organization. Data are expressed as medians with related confidence intervals. * *p* < 0.05. (**d**) Representative Western blot (WB) analysis of indicated proteins in KGN cells treated with gonadotropins (FSH, hCG, or FSH/hCG). (**d**) The histogram illustrates the relative levels of pERK (normalized to total ERK) and pROCK (normalized to tubulin). The ratios of pERK/ERK and pROCK/TUB in untreated KGN cells (CTR) are defined as 1. The results are shown as the mean ± standard deviation (SD) from three independent biological replicates (* = *p* < 0.05).

**Table 1 ijms-26-04108-t001:** Differential protein expression based on fold change: (**a**) FSH; (**b**) hCG; (**c**) FSH/hCG. (UP) Increase of expression = Log2(Fold Change) > 1.5; (DOWN) Decrease of expression = Log2(Fold Change) < −1.5.

(**a**) **FSH**			
**UP**		**DOWN**	
**2.462605**	Alpha-N-acetylglucosaminidase	**−5.50561**	Endoplasmin
**1.752348**	Microfibril-associated glycoprotein	**−4.37476**	Glucose-6-phosphate isomerase
**2.343508**	Laminin subunit beta-2	**−3.23422**	Talin-1
**1.645163**	Gamma-glutamyl hydrolase	**−2.7957**	Phosphoglucomutase-1
**1.798968**	Apolipoprotein B-100	**−3.28275**	Rab GDP dissociation inhibitor beta
**2.606067**	CD81 antigen	**−5.27955**	Elongation factor 1-alpha 1
**1.694437**	Platelet-derived growth factor D	**−4.19864**	Filamin-C
**1.65653**	Laminin subunit alpha-1	**−2.31899**	Protein S100-A11
**2.022212**	Laminin subunit alpha-5	**−2.32629**	Thioredoxin reductase 1
**2.619948**	Plasma serine protease inhibitor	**−3.1384**	14-3-3 protein theta
**1.563853**	Golgi membrane protein 1	**−2.98276**	Filamin-B
**2.426078**	Matrilin-2	**−1.56277**	Ribonuclease inhibitor
**2.406238**	Twisted gastrulation protein homolog 1	**−1.85338**	60S ribosomal protein L10a
**1.552728**	Nidogen-1	**−2.00479**	Plastin-3
**1.784398**	Pigment epithelium-derived factor	**−2.98947**	L-lactate dehydrogenase B chain
**1.824181**	Prothrombin	**−2.08653**	Neuropilin-2
**1.998121**	Pleiotrophin	**−2.60304**	Protein disulfide-isomerase
**1.537331**	Growth arrest-specific protein 6	**−4.62631**	Vinculin
**2.489509**	Hemicentin-1	**−3.09309**	Calreticulin
**2.567545**	Insulin-like growth factor-binding protein 4	**−1.90195**	Coactosin-like protein
**1.952312**	Collagen alpha-1(V) chain	**−1.88261**	Heterogeneous nuclear ribonucleoproteins A2/B1
**2.031291**	EGF-containing fibulin-like extracellular matrix protein 2	**−2.14715**	Phosphoglycerate mutase 1
**1.808948**	CD109 antigen	**−4.018**	Moesin
**2.017599**	CD59 glycoprotein	**−2.65851**	Transgelin-2
**1.713896**	Complement C1s subcomponent	**−2.37848**	Ubiquitin carboxyl-terminal hydrolase isozyme L1
**3.488902**	Lysyl oxidase homolog 1	**−1.73877**	Elongation factor 2
**1.802008**	Isoform 6 of Agrin	**−1.51905**	Di-N-acetylchitobiase
**1.88032**	Collagen alpha-1(XII) chain	**−1.8288**	Endoplasmic reticulum chaperone BiP
**1.787403**	Complement factor H	**−1.51655**	Phosphoglycerate kinase 1
**1.951293**	Fibulin-2	**−2.03801**	Filamin-A
**1.871816**	Collagen alpha-2(V) chain	**−3.03771**	Heat shock cognate 71 kDa protein
**1.548432**	Collagen alpha-1(IV) chain	**−3.49252**	Alpha-actinin-1
**1.932117**	Collagen alpha-1(III) chain	**−1.5572**	Peroxiredoxin-1
**2.011173**	Basement membrane-specific heparan sulfate proteoglycan core protein	**−2.16268**	Pyruvate kinase PKM
		**−1.90567**	Profilin-1
		**−1.76446**	Peptidyl-prolyl cis-trans isomerase A
		**−2.10461**	Cofilin-1
		**−1.60232**	14-3-3 protein zeta/delta
		**−2.30693**	14-3-3 protein epsilon
		**−2.28949**	Fructose-bisphosphate aldolase A
		**−3.05139**	Alpha-enolase
		**−2.43163**	Galectin-1
		**−2.50007**	Vimentin
		**−2.22575**	Keratin, type II cytoskeletal 1
(**b**) **hCG**			
**UP**		**DOWN**	
**1.571257**	Coatomer subunit epsilon	**−2.23692**	Actin-related protein 2
**1.927359**	Alpha-N-acetylglucosaminidase	**−1.60307**	Purine nucleoside phosphorylase
**1.89474**	Cation-independent mannose-6-phosphate receptor	**−2.79295**	Destrin
**2.054181**	Peroxidasin homolog	**−1.90296**	Zyxin
**1.580536**	Gamma-glutamyl hydrolase	**−1.52848**	Puromycin-sensitive aminopeptidase
**3.314857**	Protein CutA	**−1.90618**	Xaa-Pro dipeptidase
**2.227123**	Glia maturation factor beta	**−2.78443**	Inter-alpha-trypsin inhibitor heavy chain H1
**2.208065**	Apolipoprotein E	**−2.08102**	Proliferating cell nuclear antigen
**2.532013**	Heat shock 70 kDa protein 4	**−3.82035**	Neuropilin-2
**1.578419**	Collagen alpha-1(VIII) chain	**−2.27668**	Protein disulfide-isomerase A4
**1.601187**	Microtubule-associated protein 4	**−1.82228**	Cytochrome c
**2.059668**	Cell growth regulator with EF hand domain protein 1	**−1.74853**	Peroxiredoxin-6
**3.221587**	Serine/threonine-protein phosphatase CPPED1	**−1.97661**	Target of Nesh-SH3
**1.655928**	Matrilin-2	**−1.50756**	Adenylyl cyclase-associated protein 1
**1.663284**	Beta-hexosaminidase subunit beta	**−1.81288**	60S ribosomal protein L12
**1.599876**	Clathrin heavy chain 1	**−3.48137**	Di-N-acetylchitobiase
**2.113103**	Complement factor B	**−3.81346**	Heat shock protein HSP 90-alpha
**2.010879**	Myosin-9	**−3.23123**	Plectin
**1.582737**	Prothrombin	**−1.59051**	CD9 antigen
**1.606319**	Pigment epithelium-derived factor	**−1.63406**	Keratin, type I cytoskeletal 18
**1.586922**	Hemicentin-1	**−2.15751**	Endoplasmin
**3.040665**	DNA damage-binding protein 1	**−2.51708**	Stathmin
**3.919236**	Fatty acid synthase	**−1.55216**	Isoform 3 of Tropomyosin alpha-1 chain
**3.001855**	Keratin, type II cytoskeletal 2 epidermal	**−2.46508**	Vinculin
**1.590041**	Hemopexin	**−1.7867**	Alpha-actinin-1
		**−2.59979**	Tubulin beta chain
		**−2.55098**	Keratin, type II cytoskeletal 1
(**c**) **FSH/hCG**			
**UP**		**DOWN**	
**1.785251**	Alpha-N-acetylglucosaminidase	**−2.2124**	Secernin-1
**1.873705**	F-actin-capping protein subunit alpha-2	**−2.79808**	Xaa-Pro dipeptidase
**3.283473**	Protein CutA	**−3.22648**	Inter-alpha-trypsin inhibitor heavy chain H1
**1.913249**	Heat shock 70 kDa protein 4	**−2.36545**	Protein FAM3C
**2.007756**	Cytosolic non-specific dipeptidase	**−1.99903**	Adenosylhomocysteinase
**1.661596**	1,4-alpha-glucan-branching enzyme	**−4.44478**	Neuropilin-2
**2.652092**	Peroxidasin homolog	**−1.82006**	Zyxin
**2.316264**	Apolipoprotein B-100	**−2.95341**	Protein S100-A16
**1.710004**	Receptor of activated protein C kinase 1	**−1.85672**	Vitamin K-dependent protein S
**2.01397**	Twisted gastrulation protein homolog 1	**−1.7749**	F-actin-capping protein subunit beta
**1.562129**	Semaphorin-7A	**−1.88796**	Proliferating cell nuclear antigen
**1.994896**	Clathrin heavy chain 1	**−4.29404**	T-complex protein 1 subunit zeta
**3.373673**	Gamma-glutamyl hydrolase	**−2.26462**	Target of Nesh-SH3
**1.555267**	Integrin beta-1	**−1.67984**	Nucleolin
**2.400909**	Complement factor B	**−1.5896**	Transforming growth factor-beta
**1.849335**	Pigment epithelium-derived factor	**−1.65088**	Keratin, type I cytoskeletal 18
**2.055622**	Prothrombin	**−3.27354**	Heat shock protein HSP 90-alpha
**4.180675**	Fatty acid synthase	**−2.76699**	Plectin
**1.51762**	Afamin	**−1.74094**	Glypican-1
**1.886488**	Nucleobindin-1	**−2.35994**	Endoplasmin
**2.397546**	Gamma-enolase	**−2.17844**	Serglycin
**2.673931**	Malate dehydrogenase	**−2.99955**	Stathmin
**3.685358**	Keratin, type II cytoskeletal 2 epidermal	**−1.80893**	Complement C1r subcomponent
**2.168184**	Hemopexin	**−2.08319**	Vinculin
		**−2.8778**	Tubulin beta chain

**Table 2 ijms-26-04108-t002:** Pathways analysis based on fold change (FSH/CTR). (**a**) First 15 pathways based on 34 Up-regulated proteins; (**b**) First 15 pathways based on 44 Down-regulated proteins.

(**a**)					
**Pathway Identifier**	**Pathway Name**	**Entities Found**	**Entities Total**	**Entities pValue**	**Entities FDR**
R-HSA-1474244	Extracellular matrix organization	14	300	4.98 × 10^−14^	7.23 × 10^−12^
R-HSA-3000171	Non-integrin membrane-ECM interactions	8	59	7.12 × 10^−12^	5.12 × 10^−10^
R-HSA-3000178	ECM proteoglycans	8	76	5.21 × 10^−11^	2.50 × 10^−09^
R-HSA-3000157	Laminin interactions	6	30	3.74 × 10^−10^	1.08 × 10^−08^
R-HSA-8874081	MET activates PTK2 signaling	6	30	3.74 × 10^−10^	1.08 × 10^−08^
R-HSA-8875878	MET promotes cell motility	6	41	2.38 × 10^−09^	5.71 × 10^−08^
R-HSA-1474228	Degradation of the extracellular matrix	8	140	6.08 × 10^−09^	1.22 × 10^−07^
R-HSA-2022090	Assembly of collagen fibrils and other multimeric structures	6	61	2.48 × 10^−08^	4.46 × 10^−07^
R-HSA-6806834	Signaling by MET	6	80	1.21 × 10^−07^	1.94 × 10^−06^
R-HSA-8948216	Collagen chain trimerization	5	44	1.97 × 10^−07^	2.76 × 10^−06^
R-HSA-1474290	Collagen formation	6	90	2.41 × 10^−07^	3.13 × 10^−06^
R-HSA-186797	Signaling by PDGF	5	60	9.00 × 10^−07^	1.08 × 10^−05^
R-HSA-1442490	Collagen degradation	5	64	1.23 × 10^−06^	1.36 × 10^−05^
R-HSA-381426	Regulation of Insulin-like Growth Factor (IGF) transport and uptake by Insulin-like Growth Factor Binding Proteins (IGFBPs)	6	124	1.53 × 10^−06^	1.39 × 10^−05^
R-HSA-1650814	Collagen biosynthesis and modifying enzymes	5	67	1.54 × 10^−06^	1.39 × 10^−05^
(**b**)					
**Pathway Identifier**	**Pathway Name**	**Entities Found**	**Entities Total**	**Entities pValue**	**Entities FDR**
R-HSA-114608	Platelet degranulation	10	128	9.37 × 10^−11^	2.14 × 10^−08^
R-HSA-76005	Response to elevated platelet cytosolic Ca^2+^	10	133	1.35 × 10^−10^	2.14 × 10^−08^
R-HSA-6798695	Neutrophil degranulation	14	478	3.73 × 10^−09^	3.92 × 10^−07^
R-HSA-76002	Platelet activation, signaling and aggregation	11	265	7.06 × 10^−09^	5.58 × 10^−07^
R-HSA-70171	Glycolysis	7	80	3.62 × 10^−08^	2.28 × 10^−06^
R-HSA-168256	Immune System	25	2188	1.38 × 10^−07^	7.17 × 10^−06^
R-HSA-70326	Glucose metabolism	7	100	1.62 × 10^−07^	7.30 × 10^−06^
R-HSA-70263	Gluconeogenesis	5	35	3.42 × 10^−07^	1.33 × 10^−05^
R-HSA-9020591	Interleukin-12 signaling	5	46	1.30 × 10^−06^	4.54 × 10^−05^
R-HSA-5628897	TP53 Regulates Metabolic Genes	6	88	1.55 × 10^−06^	4.80 × 10^−05^
R-HSA-447115	Interleukin-12 family signaling	5	56	3.36 × 10^−06^	9.42 × 10^−05^
R-HSA-381183	ATF6 (ATF6-alpha) activates chaperone genes	3	10	1.00 × 10^−05^	2.50 × 10^−04^
R-HSA-168249	Innate Immune System	16	1197	1.04 × 10^−05^	2.50 × 10^−04^
R-HSA-8950505	Gene and protein expression by JAK-STAT signaling after Interleukin-12 stimulation	4	37	1.64 × 10^−05^	3.61 × 10^−04^
R-HSA-381033	ATF6 (ATF6-alpha) activates chaperones	3	12	1.72 × 10^−05^	3.62 × 10^−04^

**Table 3 ijms-26-04108-t003:** Pathways analysis based on fold change (hCG/CTR). (**a**) First 15 pathways based on 25 Up-regulated proteins; (**b**) First 15 pathways based on 27 Down-regulated proteins.

(**a**)					
**Pathway Identifier**	**Pathway Name**	**Entities Found**	**Entities Total**	**Entities pValue**	**Entities FDR**
R-HSA-199992	trans-Golgi Network Vesicle Budding	3	74	6.01 × 10^−04^	9.61 × 10^−02^
R-HSA-5653656	Vesicle-mediated transport	7	762	1.13 × 10^−03^	9.61 × 10^−02^
R-HSA-3656248	Defective HEXB causes GM2G2	1	1	2.23 × 10^−03^	9.61 × 10^−02^
R-HSA-2206282	MPS IIIB - Sanfilippo syndrome B	1	1	2.23 × 10^−03^	9.61 × 10^−02^
R-HSA-432720	Lysosome Vesicle Biogenesis	2	37	3.13 × 10^−03^	9.61 × 10^−02^
R-HSA-8964043	Plasma lipoprotein clearance	2	37	3.13 × 10^−03^	9.61 × 10^−02^
R-HSA-9024446	NR1H2 and NR1H3-mediated signaling	2	48	5.19 × 10^−03^	9.61 × 10^−02^
R-HSA-9672391	Defective F8 cleavage by thrombin	1	3	6.69 × 10^−03^	9.61 × 10^−02^
R-HSA-9657688	Defective factor XII causes hereditary angioedema	1	3	6.69 × 10^−03^	9.61 × 10^−02^
R-HSA-432722	Golgi Associated Vesicle Biogenesis	2	56	6.99 × 10^−03^	9.61 × 10^−02^
R-HSA-9725370	Signaling by ALK fusions and activated point mutants	2	57	7.23 × 10^−03^	9.61 × 10^−02^
R-HSA-9700206	Signaling by ALK in cancer	2	57	7.23 × 10^−03^	9.61 × 10^−02^
R-HSA-2022090	Assembly of collagen fibrils and other multimeric structures	2	61	8.23 × 10^−03^	9.61 × 10^−02^
R-HSA-8964026	Chylomicron clearance	1	5	1.11 × 10^−02^	9.61 × 10^−02^
R-HSA-173736	Alternative complement activation	1	5	1.11 × 10^−02^	9.61 × 10^−02^
(**b**)					
**Pathway Identifier**	**Pathway Name**	**Entities Found**	**Entities Total**	**Entities pValue**	**Entities FDR**
R-HSA-6798695	Neutrophil degranulation	8	478	8.98 × 10^−06^	2.75 × 10^−03^
R-HSA-5336415	Uptake and function of diphtheria toxin	2	7	1.26 × 10^−04^	1.85 × 10^−02^
R-HSA-114608	Platelet degranulation	4	128	2.11 × 10^−04^	1.85 × 10^−02^
R-HSA-76005	Response to elevated platelet cytosolic Ca^2+^	4	133	2.44 × 10^−04^	1.85 × 10^−02^
R-HSA-168249	Innate Immune System	9	1197	1.08 × 10^−03^	6.58 × 10^−02^
R-HSA-9013418	RHOBTB2 GTPase cycle	2	23	1.33 × 10^−03^	6.78 × 10^−02^
R-HSA-9735786	Nucleotide catabolism defects	1	1	2.32 × 10^−03^	8.48 × 10^−02^
R-HSA-9735763	Defective PNP disrupts phosphorolysis of (deoxy)guanosine and (deoxy)inosine	1	1	2.32 × 10^−03^	8.48 × 10^−02^
R-HSA-9706574	RHOBTB GTPase Cycle	2	35	3.03 × 10^−03^	8.48 × 10^−02^
R-HSA-76002	Platelet activation, signaling and aggregation	4	265	3.12 × 10^−03^	8.48 × 10^−02^
R-HSA-3299685	Detoxification of Reactive Oxygen Species	2	39	3.74 × 10^−03^	8.48 × 10^−02^
R-HSA-5339562	Uptake and actions of bacterial toxins	2	48	5.59 × 10^−03^	8.48 × 10^−02^
R-HSA-422475	Axon guidance	5	558	8.49 × 10^−03^	8.48 × 10^−02^
R-HSA-5218859	Regulated Necrosis	2	62	9.14 × 10^−03^	8.48 × 10^−02^
R-HSA-9665230	Drug resistance in ERBB2 KD mutants	1	4	9.25 × 10^−03^	8.48 × 10^−02^

**Table 4 ijms-26-04108-t004:** Pathways analysis based on fold change (FSH+hCG/CTR). (**a**) First 15 pathways based on 24 Up-regulated proteins; (**b**) First 15 pathways based on 25 Down-regulated proteins.

(**a**)					
**Pathway Identifier**	**Pathway Name**	**Entities Found**	**Entities Total**	**Entities pValue**	**Entities FDR**
R-HSA-8964038	LDL clearance	2	19	7.82 × 10^−04^	6.95 × 10^−02^
R-HSA-416700	Other semaphorin interactions	2	19	7.82 × 10^−04^	6.95 × 10^−02^
R-HSA-2206282	MPS IIIB - Sanfilippo syndrome B	1	1	2.15 × 10^−03^	6.95 × 10^−02^
R-HSA-381426	Regulation of Insulin-like Growth Factor (IGF) transport and uptake by Insulin-like Growth Factor Binding Proteins (IGFBPs)	3	124	2.34 × 10^−03^	6.95 × 10^−02^
R-HSA-5663084	Diseases of carbohydrate metabolism	2	34	2.45 × 10^−03^	6.95 × 10^−02^
R-HSA-70263	Gluconeogenesis	2	35	2.60 × 10^−03^	6.95 × 10^−02^
R-HSA-2132295	MHC class II antigen presentation	3	130	2.68 × 10^−03^	6.95 × 10^−02^
R-HSA-8964043	Plasma lipoprotein clearance	2	37	2.90 × 10^−03^	6.95 × 10^−02^
R-HSA-432720	Lysosome Vesicle Biogenesis	2	37	2.90 × 10^−03^	6.95 × 10^−02^
R-HSA-71387	Metabolism of carbohydrates	4	302	3.72 × 10^−03^	7.82 × 10^−02^
R-HSA-3878781	Glycogen storage disease type IV (GBE1)	1	3	6.43 × 10^−03^	8.84 × 10^−02^
R-HSA-9672391	Defective F8 cleavage by thrombin	1	3	6.43 × 10^−03^	8.84 × 10^−02^
R-HSA-9657688	Defective factor XII causes hereditary angioedema	1	3	6.43 × 10^−03^	8.84 × 10^−02^
R-HSA-373755	Semaphorin interactions	2	64	8.36 × 10^−03^	8.84 × 10^−02^
R-HSA-3000497	Scavenging by Class H Receptors	1	4	8.57 × 10^−03^	8.84 × 10^−02^
(**b**)					
**Pathway Identifier**	**Pathway Name**	**Entities Found**	**Entities Total**	**Entities pValue**	**Entities FDR**
R-HSA-114608	Platelet degranulation	4	128	1.55 × 10^−04^	2.70 × 10^−02^
R-HSA-76005	Response to elevated platelet cytosolic Ca^2+^	4	133	1.79 × 10^−04^	2.70 × 10^−02^
R-HSA-9013418	RHOBTB2 GTPase cycle	2	23	1.14 × 10^−03^	9.05 × 10^−02^
R-HSA-5578997	Defective AHCY causes HMAHCHD	1	1	2.15 × 10^−03^	9.05 × 10^−02^
R-HSA-76002	Platelet activation, signaling and aggregation	4	265	2.33 × 10^−03^	9.05 × 10^−02^
R-HSA-9706574	RHOBTB GTPase Cycle	2	35	2.60 × 10^−03^	9.05 × 10^−02^
R-HSA-109582	Hemostasis	6	726	3.74 × 10^−03^	9.05 × 10^−02^
R-HSA-3371497	HSP90 chaperone cycle for steroid hormone receptors (SHR) in the presence of ligand	2	57	6.69 × 10^−03^	9.05 × 10^−02^
R-HSA-9665230	Drug resistance in ERBB2 KD mutants	1	4	8.57 × 10^−03^	9.05 × 10^−02^
R-HSA-9652282	Drug-mediated inhibition of ERBB2 signaling	1	4	8.57 × 10^−03^	9.05 × 10^−02^
R-HSA-9665233	Resistance of ERBB2 KD mutants to trastuzumab	1	4	8.57 × 10^−03^	9.05 × 10^−02^
R-HSA-9665737	Drug resistance in ERBB2 TMD/JMD mutants	1	4	8.57 × 10^−03^	9.05 × 10^−02^
R-HSA-9665244	Resistance of ERBB2 KD mutants to sapitinib	1	4	8.57 × 10^−03^	9.05 × 10^−02^
R-HSA-9665250	Resistance of ERBB2 KD mutants to AEE788	1	4	8.57 × 10^−03^	9.05 × 10^−02^
R-HSA-9665249	Resistance of ERBB2 KD mutants to afatinib	1	4	8.57 × 10^−03^	9.05 × 10^−02^

## Data Availability

The data presented in this study are available in this article (and Appendix A).

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
