# Peer review of "Secretory Profile Analysis of Human Granulosa Cell Line Following Gonadotropin Stimulation"

_ijms, 2025, doi:10.3390/ijms26094108_

Round 1
Reviewer 1 Report
Comments and Suggestions for Authors
The study explores the impact of gonadotropins (FSH and hCG) on human granulosa cell (GC) function using a proteomic approach. The authors analyze the secretome of KGN cells (a human granulosa-like tumor cell line) to identify differentially secreted proteins in response to hormonal stimulation. The study emphasizes cytoskeletal remodeling’s role in GC differentiation. Furthermore, key pathways, including Semaphorin 7A signaling, focal adhesion kinase (FAK), and Rho-associated protein kinase (ROCK), are highlighted as critical mediators of follicle development.
While the study provides a proteomic overview, some methodological limitations and interpretative gaps reduce its impact..
- For Fig.1 The study identifies differentially abundant proteins after gonadotropin treatment but does not correlate these changes with mRNA expression levels. While protein and transcript levels do not always align due to post-translational modifications, integrating public RNA-seq data would provide additional confidence in the findings. Without transcriptomic validation, it is unclear whether the observed proteomic changes result from transcriptional regulation, post-translational modifications, or differences in protein stability. More importantly, this comparison would also enhance the reliability of these proteomic data.
- For Fig.3, the quality of the WB will be a concern, since the background of Sema7A and Actin is very different.
- For Fig4d, In the FSH and FSH/hCG treatment groups, the TUB levels appear lower compared to the control, raising concerns about whether this is due to actual biological effects or unequal protein loading.
Author Response
Comments and Suggestions for Authors
The study explores the impact of gonadotropins (FSH and hCG) on human granulosa cell (GC) function using a proteomic approach. The authors analyze the secretome of KGN cells (a human granulosa-like tumor cell line) to identify differentially secreted proteins in response to hormonal stimulation. The study emphasizes cytoskeletal remodeling’s role in GC differentiation. Furthermore, key pathways, including Semaphorin 7A signaling, focal adhesion kinase (FAK), and Rho-associated protein kinase (ROCK), are highlighted as critical mediators of follicle development.
While the study provides a proteomic overview, some methodological limitations and interpretative gaps reduce its impact..
For Fig.1 The study identifies differentially abundant proteins after gonadotropin treatment but does not correlate these changes with mRNA expression levels. While protein and transcript levels do not always align due to post-translational modifications, integrating public RNA-seq data would provide additional confidence in the findings. Without transcriptomic validation, it is unclear whether the observed proteomic changes result from transcriptional regulation, post-translational modifications, or differences in protein stability. More importantly, this comparison would also enhance the reliability of these proteomic data.
Thank you for your observation. We agree that the comparison with mRNA expression levels and proteomic data would enhance the reliability of our data. Therefore, the principal aim of this research was to highlight “in vitro” a pattern of protein expression modulated by gonadotropins in order to identify markers of positive response to treatment. This would allow us to translate the panel of identified protein in clinical practice by measuring in serum samples of women undergoing ovarian stimulation (COS) protocols.
For Fig.3, the quality of the WB will be a concern, since the background of Sema7A and Actin is very different.
We acknowledge your comment and have modified Figure 3 to present Sema7 and Actin with identical exposure settings.
For Fig4d, In the FSH and FSH/hCG treatment groups, the TUB levels appear lower compared to the control,
raising concerns about whether this is due to actual biological effects or unequal protein loading.
Thank you for your observation. The difference is likely due to varying protein loading, but regardless, the result is presented as the ratio between the protein of interest (pROCK) and its loading control.
Reviewer 2 Report
Comments and Suggestions for Authors
In present work, Mancini et al. try to study the secretory profile of human granulosa cell line following gonadotropin stimulation. This paper showed a significant modulation of proteins involved in extracellular matrix organization and interactions, steroidogenesis and cytoskeleton remodeling. In addition, there was a dynamic upregulation of the Semaphorins pathway and, in particular, an upregulation of Semaphorin 7A following FSH/hCG combined treatment. However, there are some questions that should be explained.
Major concerns
- In clinical practice, hCG is used to mimic LH. However, hCG does not have the activity of LH, but also has a little of activity of FSH. I do not think that hCG can completely replace LH for the experimental aim.
- In mammals, the development and maturation of oocyte into growing follicle depends on ovarian granulosa cells differentiation in response to gonadotropin stimulation, including FSH and LH. However, it is a dynamic procedure for FSH and LH stimulating the development and maturation of oocyte. The early development of oocyte is mainly depends on FSH, but the late development and maturation of oocyte is mainly depends on LH, which should be considered in this study.
- LH is not only involved in ovulation, but also participates in the late development and maturation of oocyte. In vivo, the ovarian granulosa cells are under the both effects of FSH and LH. Therefore, if it is reasonable that the ovarian granulosa cells were under the treatment by FSH, hCG, or FSH/ hCG? hCG has a little of activity of FSH, and normal concentration of LH participates in the late development and maturation of oocyte, but high concentration of LH is related to ovulation.
- English grammar and writing should be checked and revised throughout the manuscript.
Minor concerns
- Abstract section should be rewritten, which should summary whole manuscript, there are too many background knowledge.
- Line 26, as a scientific paper, it should be written in the third person. There are so many ‘we’. Please check these throughout this manuscript.
- Line 45, change ‘GCs’ to ‘GC’.
- At the end of Introduction section, hypothesis and aim of this study should be included.
- Tables should be in an editable format.
- Figure 4, bars should be added in a and b.
- Lines 337-338, revise ‘a protease inhibitor’.
- Line 460, revise ‘CO2/95% O2’.
- Western blot subsection, the catalogue number of antibody should be added.
- The reference format is not consistent. Journal name of some references are in abbreviation, but some references are not. Please check these throughout Reference section.
The English could be improved to more clearly express the research.
Author Response
In present work, Mancini et al. try to study the secretory profile of human granulosa cell line following gonadotropin stimulation. This paper showed a significant modulation of proteins involved in extracellular matrix organization and interactions, steroidogenesis and cytoskeleton remodeling. In addition, there was a dynamic upregulation of the Semaphorins pathway and, in particular, an upregulation of Semaphorin 7A following FSH/hCG combined treatment. However, there are some questions that should be explained.
Major concerns
- In clinical practice, hCG is used to mimic LH. However, hCG does not have the activity of LH, but also has a little of activity of FSH. I do not think that hCG can completely replace LH for the experimental aim.
- In mammals, the development and maturation of oocyte into growing follicle depends on ovarian granulosa cells differentiation in response to gonadotropin stimulation, including FSH and LH. However, it is a dynamic procedure for FSH and LH stimulating the development and maturation of oocyte. The early development of oocyte is mainly depends on FSH, but the late development and maturation of oocyte is mainly depends on LH, which should be considered in this study.
- LH is not only involved in ovulation, but also participates in the late development and maturation of oocyte. In vivo, the ovarian granulosa cells are under the both effects of FSH and LH. Therefore, if it is reasonable that the ovarian granulosa cells were under the treatment by FSH, hCG, or FSH/ hCG? hCG has a little of activity of FSH, and normal concentration of LH participates in the late development and maturation of oocyte, but high concentration of LH is related to ovulation.
Thank you for the comments. Albeit hCG cannot completely replace LH, we used it because we need to mimic in vitro, as possible, the controlled ovarian stimulation protocols. Moreover, due to experimental constraints, specifically the limited cell viability in serum-free medium, we had to apply the two gonadotropins simultaneously, rather than sequentially, which would have been more physiologically relevant.
Furthermore, this experimental approach, single gonadotropin treatments versus their combination, was selected to maximize the hormonal response, enabling consistent data collection for both single and combined treatments.
- English grammar and writing should be checked and revised throughout the manuscript.
Thank you for your comment, we revised English language in the manuscript.
Minor concerns
- Abstract section should be rewritten, which should summary whole manuscript, there are too many background knowledge.
Thank you for your comment, we rewrite the abstract according to yours suggestion.
- Line 26, as a scientific paper, it should be written in the third person. There are so many ‘we’. Please check these throughout this manuscript.
Thank you for your suggestion, we have made this change in the abstract.
- Line 45, change ‘GCs’ to ‘GC’.
Thank you, we revised it
- At the end of Introduction section, hypothesis and aim of this study should be included.
- Thank you for the comment, we modified the introduction section.
- Tables should be in an editable format.
Thank you for your suggestion, our proteomic facility provides us tables in PDF format.
- Figure 4, bars should be added in a and b.
Thank you for your suggestion, we added scale bar in figure 4(a). In figure4 (b) legend we added the information that for each image several squared ROI (100 × 100 pixels) were selected in correspondence of the inner part of the cells.
- Lines 337-338, revise ‘a protease inhibitor’.
Thank you, we revised it.
- Line 460, revise ‘CO2/95% O2’.
Thank you, we revised it.
- Western blot subsection, the catalogue number of antibody should be added.
Thank you, we revised it.
- The reference format is not consistent. Journal name of some references are in abbreviation, but some references are not. Please check these throughout Reference section.
Thank you, we revised it.
- Comments on the Quality of English Language.The English could be improved to more clearly express the research.
Thank you, we revised the English Language

Reviewer 3 Report
Comments and Suggestions for Authors
In this manuscript, the authors investigate secreted proteins and cytoskeletal changes in an immortalized human granulosa cell line. The treatments were: control, FSH, hCG, and FSH/hCG. Treatment with gonadotropins altered the profile of secreted proteins, surprisingly reducing the expression of many. Cytoskeleton changes were assessed with immune fluorescent and western blot of key enzymes. Findings indicate clear changes in cytoskeleton associated with changes in phosphorylation of FAK. Although the work is interesting a few weaknesses should be addressed.
- For proteomic data, was there any replication in data? Multiple plates being treated? Seems like some type of replication would be ideal.
- The comparison of secreted genes following treatment compared to control as presented in figure 1a,b,c and tables, but a comparison of altered proteins between FSH, hCG, and FSH/hCG would significantly improve the impact of this work. For example, is the effect of FSH/hCG an additive effect of both hormones?
- The data are presented in a seemingly odd order: I would think that the table 1 a,b,c would follow figure 1, then figure 2, then tables 2,3,4, then figure 4. Figure 3 could come after table 4 or after table 1.
- Justification for doses of FSH and hCG was not provided, please add this important information.
- Figure 4D, bar charts are missing labels for treatment groups. Minor point, but figure legend refers to these as histograms, which they are not, these are bar charts.
- Line 124: Cumulus-ovocyte should be cumulus-oocyte
- Line 70: Progestins are also often used in ovarian stimulation protocols to prevent ovulation
- Line 83: italicize “in vitro”
Comments on the Quality of English Language
Mostly good. Introduction is great, results section could use some help.
Author Response
In this manuscript, the authors investigate secreted proteins and cytoskeletal changes in an immortalized human granulosa cell line. The treatments were: control, FSH, hCG, and FSH/hCG. Treatment with gonadotropins altered the profile of secreted proteins, surprisingly reducing the expression of many. Cytoskeleton changes were assessed with immune fluorescent and western blot of key enzymes. Findings indicate clear changes in cytoskeleton associated with changes in phosphorylation of FAK. Although the work is interesting a few weaknesses should be addressed.
- For proteomic data, was there any replication in data? Multiple plates being treated? Seems like some type of replication would be ideal.
Thank you for the comment, we apologize for the omitted data, we have included it in the Material and Method section
- The comparison of secreted genes following treatment compared to control as presented in figure 1a,b,c and tables, but a comparison of altered proteins between FSH, hCG, and FSH/hCG would significantly improve the impact of this work. For example, is the effect of FSH/hCG an additive effect of both hormones?
Thank you for the comment, we added differential expressed protein analysis of FSH vs. FSH/hCG and hCG vs. FSH/hCG in the supplementary Table 2S and 3S
- The data are presented in a seemingly odd order: I would think that the table 1 a,b,c would follow figure 1, then figure 2, then tables 2,3,4, then figure 4. Figure 3 could come after table 4 or after table 1.
Thank you for the comment, figures and tables are integrated into the text, appearing in the order they are referenced within the results and discussion sections.
- Justification for doses of FSH and hCG was not provided, please add this important information.
Thank you for the comment; we added this information in Material and Method section
(ref 93-94)
- Figure 4D, bar charts are missing labels for treatment groups. Minor point, but figure legend refers to these as histograms, which they are not, these are bar charts.
Thank you for your observation. We acknowledge the mistake and have updated Figure 4D accordingly.
- Line 124: Cumulus-ovocyte should be cumulus-oocyte
- Line 70: Progestins are also often used in ovarian stimulation protocols to prevent ovulation
- Line 83: italicize “in vitro”
Thank you for your suggestions, we modified cumulus-ovocyte in cumulus-oocyte, and we italicized “in vitro”. Furthermore, we added in the introduction that Progestins are also often used in ovarian stimulation protocols to prevent ovulation (ref 12).
We conducted a final revision of the English language within the results and discussion sections.

Round 2
Reviewer 1 Report
Comments and Suggestions for Authors
The author did not address my previous Question 1, which pertains to the quality of the results.
Author Response
For Fig.1 The study identifies differentially abundant proteins after gonadotropin treatment but does not correlate these changes with mRNA expression levels. While protein and transcript levels do not always align due to post-translational modifications, integrating public RNA-seq data would provide additional confidence in the findings. Without transcriptomic validation, it is unclear whether the observed proteomic changes result from transcriptional regulation, post-translational modifications, or differences in protein stability. More importantly, this comparison would also enhance the reliability of these proteomic data.
We appreciate your insightful suggestion to incorporate public RNA-seq data. This integration holds significant potential to enhance our study's findings by providing essential context for the observed proteomic alterations and offering more profound understanding of the underlying regulatory processes. Observing corresponding changes in mRNA levels (in the same direction) for a substantial number of the differentially abundant proteins would considerably increase our confidence that transcriptional regulation contributes, at least partially, to the proteomic changes, thus offering a more robust biological explanation for our findings. The practicality of this suggestion hinges on the existence of pertinent public RNA-seq datasets, ideally derived from similar cell types (KGN ovarian granulosa cells) treated with comparable gonadotropins (FSH, and hCG alone or in combination) under analogous experimental conditions. The work of Tremblay P. and Sirard M*., who employed RNA-seq to investigate key intracellular signalling pathways induced by FSH in KGN granulosa cells, is of interest. They found a modulation of differentiation and steroidogenesis pathways upon FSH treatment, providing additional confidence to our data. Regardless, we have added a description of their RNA-seq data discussion section.
* Tremblay,P.G.; Sirard, M-A. Gene analysis of major signaling pathways regulated by gonadotropins in human ovarian Granulosa tumor cells (KGN). Biol Reprod ,2020, 103(3), 583–598. doi:10.1093/biolre/ioaa079
(The data discussed in this publication have been deposited in NCBI’s Gene Expression Omnibus (GEO) and are accessible through GEO Series accession number GSE137608 (https://www.ncbi.nlm.nih.gov/ geo/query/acc.cgi?acc=GSE137608).
Reviewer 2 Report
Comments and Suggestions for Authors
Thanks for author’s responses. However, there are STILL some questions that should be explained.
Major concerns (authors should response three questions one by one)
- In clinical practice, hCG is used to mimic LH. However, hCG does not have the activity of LH, but also has a little of activity of FSH. I do not think that hCG can completely replace LH for the experimental aim.
- In mammals, the development and maturation of oocyte into growing follicle depends on ovarian granulosa cells differentiation in response to gonadotropin stimulation, including FSH and LH. However, it is a dynamic procedure for FSH and LH stimulating the development and maturation of oocyte. The early development of oocyte is mainly depends on FSH, but the late development and maturation of oocyte is mainly depends on LH, which should be considered in this study.
- LH is not only involved in ovulation, but also participates in the late development and maturation of oocyte. In vivo, the ovarian granulosa cells are under the both effects of FSH and LH. Therefore, if it is reasonable that the ovarian granulosa cells were under the treatment by FSH, hCG, or FSH/ hCG? hCG has a little of activity of FSH, and normal concentration of LH participates in the late development and maturation of oocyte, but high concentration of LH is related to ovulation.
Minor concerns
- English grammar and writing STILL should be checked and revised throughout the manuscript.
Line 92, revise ‘(ROCK-1 and -2).FAK’.
Figures 3 and 4, using yellow background is not suitable.
Line 476, revise ‘antibiotics.. The’.
- The reference format is STILL need to check. There are so many wrong, so this reviewer doubt if the authors really read these papers. Authors should check these one by one.
For example,
Ref 16, ‘Cochrane Database of Systematic Reviews 2017, Issue 3. Art. No.: CD012586.’.
Ref 24, ‘JCS, 2002’.
Ref 31, ‘-BBA - Molecular Basis’.
Ref 34, ‘MBoC, 2002’.
……
Comments on the Quality of English LanguageThe English could be improved to more clearly express the research.
Author Response
2) Thanks for author’s responses. However, there are STILL some questions that should be explained.
Major concerns (authors should response three questions one by one)
- In clinical practice, hCG is used to mimic LH. However, hCG does not have the activity of LH, but also has a little of activity of FSH. I do not think that hCG can completely replace LH for the experimental aim.
We appreciate the comment provided. Clinically, hCG is a well-established mimic of LH, especially for ovulation induction and luteal phase support. However, it is important to recognize that hCG does not possess identical activity to LH. While it binds to the shared LH/hCG receptor, its extended half-life and subtle variations in signaling pathways can lead to differing physiological outcomes compared to the pulsatile secretion of endogenous LH. Furthermore, hCG exhibits a weak capacity to bind to and stimulate the FSH receptor, albeit with a significantly reduced affinity compared to FSH itself. When choosing between LH and hCG, careful consideration of each experiment's specific goals and the potential influence of their differences is crucial. Our study, however, focused on identifying markers of positive response to gonadotropin treatment by replicating controlled ovarian stimulation (COS) protocols involving FSH and hCG.
- In mammals, the development and maturation of oocyte into growing follicle depends on ovarian granulosa cells differentiation in response to gonadotropin stimulation, including FSH and LH. However, it is a dynamic procedure for FSH and LH stimulating the development and maturation of oocyte. The early development of oocyte is mainly depends on FSH, but the late development and maturation of oocyte is mainly depends on LH, which should be considered in this study.
The communication between the gonadotropins (FSH and LH) and the oocyte is largely mediated by the surrounding granulosa cells (GC). These cells undergo crucial differentiation in response to FSH and LH stimulation, creating the necessary microenvironment for oocyte growth and maturation. FSH plays a primary role in the initial stages of follicular development stimulating GC proliferation and inducing the expression of key enzymes involved in estrogen synthesis (e.g., aromatase). The oocyte itself grows during this phase, and its development is strongly influenced by paracrine factors secreted by FSH-stimulated granulosa cells. As follicles mature and reach the preovulatory stage, the role of LH becomes increasingly important. The LH surge is the critical signal for the final stages of oocyte maturation, including resumption of meiosis, cumulus expansion, steroidogenesis, and ovulation. In our in vitro model, experimental constraints, specifically the time –dependent reduction of cell viability observed in serum-free medium, necessitated the simultaneous administration of both gonadotropins. However, our experimental design, which included treatments with individual gonadotropins alongside their combination, was implemented to maximize hormonal responses and facilitate consistent data acquisition for all treatment groups.
- LH is not only involved in ovulation, but also participates in the late development and maturation of oocyte. In vivo, the ovarian granulosa cells are under the both effects of FSH and LH. Therefore, if it is reasonable that the ovarian granulosa cells were under the treatment by FSH, hCG, or FSH/ hCG? hCG has a little of activity of FSH, and normal concentration of LH participates in the late development and maturation of oocyte, but high concentration of LH is related to ovulation.
As you correctly highlight, ovarian granulosa cells in vivo encounter both FSH and LH, particularly during follicle maturation. This co-stimulation likely results in intricate and synergistic impacts on their differentiation and function. Therefore, we considered the following treatment groups for in vitro studies on KGN cells:
- FSH alone: To examine its specific effects, especially those pertinent to early follicular development, granulosa cell proliferation, and initial differentiation.
- hCG alone: To primarily simulate LH receptor activation and study its signalling effects in later stages.
- FSH/hCG combination: To best replicate the in vivo environment during late follicular development, enabling the study of potential synergistic effects of FSH and LH signalling on granulosa cell function and differentiation.
We acknowledge that a limitation of this study is the inability to administer the two gonadotropins sequentially. However, as previously stated, experimental constraints necessitated this protocol, which still enabled us to highlight the pathways modulated by the combined treatment and subsequently identify markers of positive response to gonadotropin treatment.
Minor concerns
- English grammar and writing STILL should be checked and revised throughout the manuscript.
Thank you we checked English grammar.
Line 92, revise ‘(ROCK-1 and -2).FAK’.
Thank you we revised it.
Figures 3 and 4, using yellow background is not suitable.
Thank you, yellow background was made in the revised paper submission to highlight corrections that occurred in the first round review. Surely it will be removed in the final version.
Line 476, revise ‘antibiotics.. The’.
Thank you we revised it.
- The reference format is STILL need to check. There are so many wrong, so this reviewer doubt if the authors really read these papers. Authors should check these one by one.
For example,
Ref 16, ‘Cochrane Database of Systematic Reviews 2017, Issue 3. Art. No.: CD012586.’.
Ref 24, ‘JCS, 2002’.
Ref 31, ‘-BBA - Molecular Basis’.
Ref 34, ‘MBoC, 2002’.
Thank you we checked and revised one by one. Ref 16
Round 3
Reviewer 1 Report
Comments and Suggestions for Authors
The previous question remains unaddressed.
While the author included some statements regarding related RNA-seq data, it remains unclear how the presented proteomic data correlate with publicly available transcriptomic datasets. Importantly, identifying consistent patterns would strengthen the reliability of the proteomic results, while discrepancies could provide valuable insights into post-transcriptional regulation.
Author Response
- The previous question remains unaddressed.
While the author included some statements regarding related RNA-seq data, it remains unclear how the presented proteomic data correlate with publicly available transcriptomic datasets. Importantly, identifying consistent patterns would strengthen the reliability of the proteomic results, while discrepancies could provide valuable insights into post-transcriptional regulation.
Thank you for your comment. We apologize that our answer was not fully comprehensive.
As you mentioned we have already included some statement regarding public RNA-seq data although they concern only FSH treatment. In literature, we did not found other public RNA data that resemble our experimental model. The present study analyses secreted proteins known to be extensively regulated at both transcriptional and post-transcriptional levels. While the lack of transcriptional data is a significant limitation, the primary aim of this work remains the identification of protein markers for positive response to gonadotropin treatment, aiming to translate these findings into clinical practice for COS protocols. We include a comment about this limitation at the end of discussion section.
Reviewer 2 Report
Comments and Suggestions for Authors
Thanks for author’s responses. However, there are STILL some questions that should be explained.
- As this reviewer reviews in the previous two reports that hCG does not have the activity of LH, but also has a little of activity of FSH, and LH is not only involved in ovulation, but also participates in the late development and maturation of oocyte. In addition, authors have acknowledged the limitation of this study. Therefore, these should be included in the Discussion section.
- Scientific paper should be written in a third person manner. There are many ‘we’, which should be corrected.
- Percent match is 51%, which is very higher.
The English could be improved to more clearly express the research.
Author Response
1. As this reviewer reviews in the previous two reports that hCG does not have the activity of LH, but also has a little of activity of FSH, and LH is not only involved in ovulation, but also participates in the late development and maturation of oocyte. In addition, authors have acknowledged the limitation of this study. Therefore, these should be included in the Discussion section.
Thank you. As you suggest we included the comment about hCG activity at the beginning of the discussion section.
2.Scientific paper should be written in a third person manner. There are many ‘we’, which should be corrected.
Thank you, we corrected the manuscript according to your suggestion.
2. Percent match is 51%, which is very higher.
Thank you, the plagiarism was revised.
Round 4
Reviewer 1 Report
Comments and Suggestions for Authors
I have no further questions.